# High-performance enrichment-based genome sequencing to support the investigation of hepatitis A virus outbreaks

Sara E. Zufan,[1,2] Karolina Mercoulia,[2,3] Jason C. Kwong,[4,5] Louise M. Judd,[2,6] Benjamin P. Howden,[1,2,3] Torsten Seemann,[1,2] Timothy P. Stinear[1,2]

**ABSTRACT** Hepatitis A virus (HAV) infections are an increasing public health concern in low-endemicity regions due to outbreaks from food-borne infections and sustained transmission among vulnerable groups, including persons experiencing homelessness, those who inject drugs, and men who have sex with men, which is further compounded by aging, unvaccinated populations. DNA sequence characterization of HAV for source tracking is performed by comparing small subgenomic regions of the virus. While this approach has been successful when robust epidemiological data are available, poor genetic resolution can lead to the conflation of outbreaks with sporadic cases. HAV outbreak investigations would greatly benefit from the additional phylogenetic resolution obtained by whole virus genome sequence comparisons. However, HAV genomic approaches can be difficult because of challenges in isolating the virus, low sensitivity of direct metagenomic sequencing in complex sample matrices like various foods, such as fruits, vegetables, and molluscs, and difficulty designing highly multiplexed PCR primers across diverse HAV genotypes. Here, we introduce a proof-of-concept pan-HAV oligonucleotide hybrid capture enrichment assay from serum and frozen berry specimens that yields complete and near-complete HAV genomes from as few as four input HAV genome copies. We used this method to recover HAV genomes from human serum specimens with high Cτ values (34.7–42.7), with high assay performance for all six human HAV subgenotypes, both contemporary and historical. Our approach provides a highly sensitive and streamlined workflow for HAV whole-genome sequencing from diverse sample types that can be the basis for harmonized and high-resolution molecular epidemiology during HAV outbreak surveillance.

**IMPORTANCE** This proof-of-concept study introduces a hybrid capture oligo panel for whole-genome sequencing of all six human pathogenic hepatitis A virus (HAV) subgenotypes, exhibiting a higher sensitivity than some conventional genotyping assays. The ability of hybrid capture to enrich multiple targets allows for a single, streamlined workflow, thus facilitating the potential harmonization of molecular surveillance of HAV with other enteric viruses. Even challenging sample matrices can be accommodated, making them suitable for broad implementation in clinical and public health laboratories. This innovative approach has significant implications for enhancing multijurisdictional outbreak investigations as well as our understanding of the global diversity and transmission dynamics of HAV.

**KEYWORDS** public health genomics, culture-independent diagnostics, whole-genome sequencing, food-borne surveillance, molecular epidemiology

Address correspondence to Sara E. Zufan, s.zufan@unimelb.edu.au.

The authors declare no conflict of interest.

See the funding table on p. 12.

Hepatitis A is a vaccine-preventable disease caused by hepatitis A virus (HAV), a nonenveloped virus, with a small (7.5 kb) positive-sense RNA genome in the family *Picornaviridae*. HAV is transmitted fecal-orally by consuming contaminated food or drink,

as well as person-to-person between close contacts (1). It is the leading cause of acute viral hepatitis, and symptoms can be prolonged but rarely result in fulminant hepatitis. Endemicity can be classified as high, intermediate, low, or very low and is generally associated with socioeconomic indicators such as hygienic and sanitary conditions (2). The incidence of HAV infections declined in many low-endemicity countries after the implementation of routine childhood immunization by the early 2000s (3, 4). As a result, immunity in adult populations has declined significantly, thereby increasing the number of susceptible individuals who are more likely to experience and report symptoms of hepatitis (5). Epidemiology in these regions then shifted from cyclical community-wide outbreaks associated with childcare facilities or households with children to sporadic outbreaks stemming from travelers or consumption of contaminated food from high- and intermediate-endemicity regions (2, 6–9). Another epidemiological shift began in 2016 when several North American and European countries observed a sharp increase in cases driven by person-to-person transmission, particularly in vulnerable populations including persons reporting injection drug use, experiencing homelessness, and men who have sex with men (10–19). The combination of infection in older adults and persons with one or more comorbidities has resulted in an uptick in hospitalizations and deaths related to HAV infection (5, 13, 20, 21).

Human pathogenic HAV can be characterized into three genotypes (I, II, and II), each with two subgenotypes (A and B). The distribution of subgenotypes varies geographically, although IA accounts for more than half of all reported cases (22). Combined with epidemiological data, subgenotyping has been shown to improve outbreak characterization and response by differentiating between sporadic and outbreak cases, determining the source of infection, and tracking chains of transmission (17, 18). Several genome regions are used for HAV genotyping. The most common regions in HAV query databases are found in or around a 168-nucleotide (nt) fragment of the VP1/2A junction (22–24). However, the accuracy of phylogenetic relationships inferred using these small regions has been put into question by several studies that found significant variations in genetic relatedness when comparing different subgenomic regions (25–27).

Whole-genome sequencing (WGS) comparisons have become increasingly employed for viral surveillance and outbreak detection as variation across the entire virus genome can increase the spatial and temporal phylogenetic resolution of molecular epidemiology (28). Indeed, Vaughn et al. found that subgenotyping could yield misleading transmission interpretations when applied to cases with unknown epidemiological associations, demonstrating that WGS was able to differentiate between sporadic and outbreak cases with identical VP1/2A sequences (27). This additional resolution could be particularly useful for guiding response to growing person-to-person transmission. For example, a recent analysis of a large outbreak in New York (USA) found a cluster of identical VP1/2A sequences from neighboring jurisdictions. WGS revealed 3–41 single nucleotide variants (SNVs) between cases, where the case with the greatest nucleotide difference had traveled to Florida, suggesting the infection was acquired outside of the local outbreak (29, 30).

Despite the potential benefits, WGS of HAV in a public health context has been challenging for several reasons. First, HAV grows poorly in cell culture, making isolation infeasible for routine culture-based WGS. Second, despite moderate genetic diversity compared to other RNA viruses, enough heterogeneity across the genome exists to make it difficult to design primers for multiplex amplicon sequencing (30, 31). Recently, two primer schemes for HAV WGS have been designed targeting IA and/or IB, where an effort to incorporate additional subgenotypes reduced primer efficiency (29, 32). Both studies had difficulty reliably recovering whole genomes from serum samples above 30 Cτ.

Alternatively, hybridization capture is a highly flexible option for culture-agnostic WGS. Unlike PCR-based amplification, hybrid capture reactions can contain hundreds of thousands of oligonucleotide baits (probes) without interactions and are unaffected by amplification inhibitors in complex sample matrices. Furthermore, conservative tiling probe design can improve the ability to capture novel sequences as well as

increase sensitivity in degraded samples. These characteristics make it an ideal solution for routine WGS in public health laboratories for pathogens like HAV that are genetically diverse and present in a variety of complex sample matrices. The flexibility of probe design can also be leveraged to incorporate additional enteric pathogens, which can streamline library preparation and increase the sequence throughput of routine diagnostic or surveillance workflows.

Here, we present a hybridization capture panel for undertaking WGS designed to be highly sensitive across all HAV human subgenotypes. We demonstrate the sensitivity of the capture panel to recover WGS from food (berry) and human serum samples spiked with serially diluted HAV RNA ranging from 27.9 copies/µL to below the limit of detection (LOD; <0.2 copies/µL), corresponding to cycle threshold (Cτ) values 33.8 to >42.0. We also used the panel to recover complete and near-complete (93.3%) genomes from clinical serum samples of infected patients, with and without subgenotyping data available. Finally, we used *in silico* prediction to extrapolate the efficiency of the panel to capture subgenotypes not evaluated in this study.

## MATERIALS AND METHODS

### Pan-HAV probe design

Complete genomes of human pathogenic HAV were downloaded from GenBank on 25 January 2021 ($N$ = 164). Genomes were manually selected for temporal and geographic representation of each subgenotype ($N$ = 44). Probes were designed by IDT as previously described using a 120-mer 1× tiling approach (33).

### Optimization experiment

#### *Sample preparation*

Food samples were prepared with frozen mixed berries according to International Organization for Standardization report ISO 15216-1:2017 with two modifications (34). First, RNA extraction was performed on a QIASymphony with a custom protocol using the DSP Virus/Pathogen Kit with an input volume of 800 µL and an elution volume of 60 µL. The second modification was the use of an Applied Biosystems 7500 Fast DX system for detection and quantification. "Low," "mid," and "high" virus-spiked berry specimens were prepared based on ISO 15216-2:-2019 validation studies (35). For serum samples, ultraconcentrated spike-in material was prepared by resuspending one LENTICULE disc (RM000HAV) in 222 µL PBS, followed by $10^{-1}$ serial dilutions in human sera. RNA extractions were again performed using a QiaSymphony using the DSP Virus/Pathogen Kit with an input volume of 200 µL and elution volume of 60 µL. Quantification of target copies per microliter of sample RNA was performed as described in ISO 15216-1:2017. HAV LENTICULE discs were used for both the quantification standards and as the spike-in material.

The volume of RNA extracts was equally divided to create four experimental variables: pre-enrichment, pre-enrichment with concentration, post-enrichment, and post-enrichment with concentration. A subset of samples were concentrated twofold using the Zymo Clean and Concentrate −5 kit including on-column DNase treatment with a final elution volume of 10 µL. An overview of the laboratory workflows is illustrated in Fig. 1A.

#### *Constructing sequencing libraries*

Sequencing libraries were prepared by Illumina RNA Prep with Enrichment with 8.5 µL RNA input, according to the manufacturer's protocol. Total RNA for all samples was below the recommended 10 ng. Briefly, RNA is synthesized to cDNA, followed by tagmentation library preparation. The indexed library is then hybridized with the custom probe panel, captured, and, finally, PCR amplified. Pre-enrichment controls were removed from the workflow after purification of the indexed library. Enriched sample libraries were

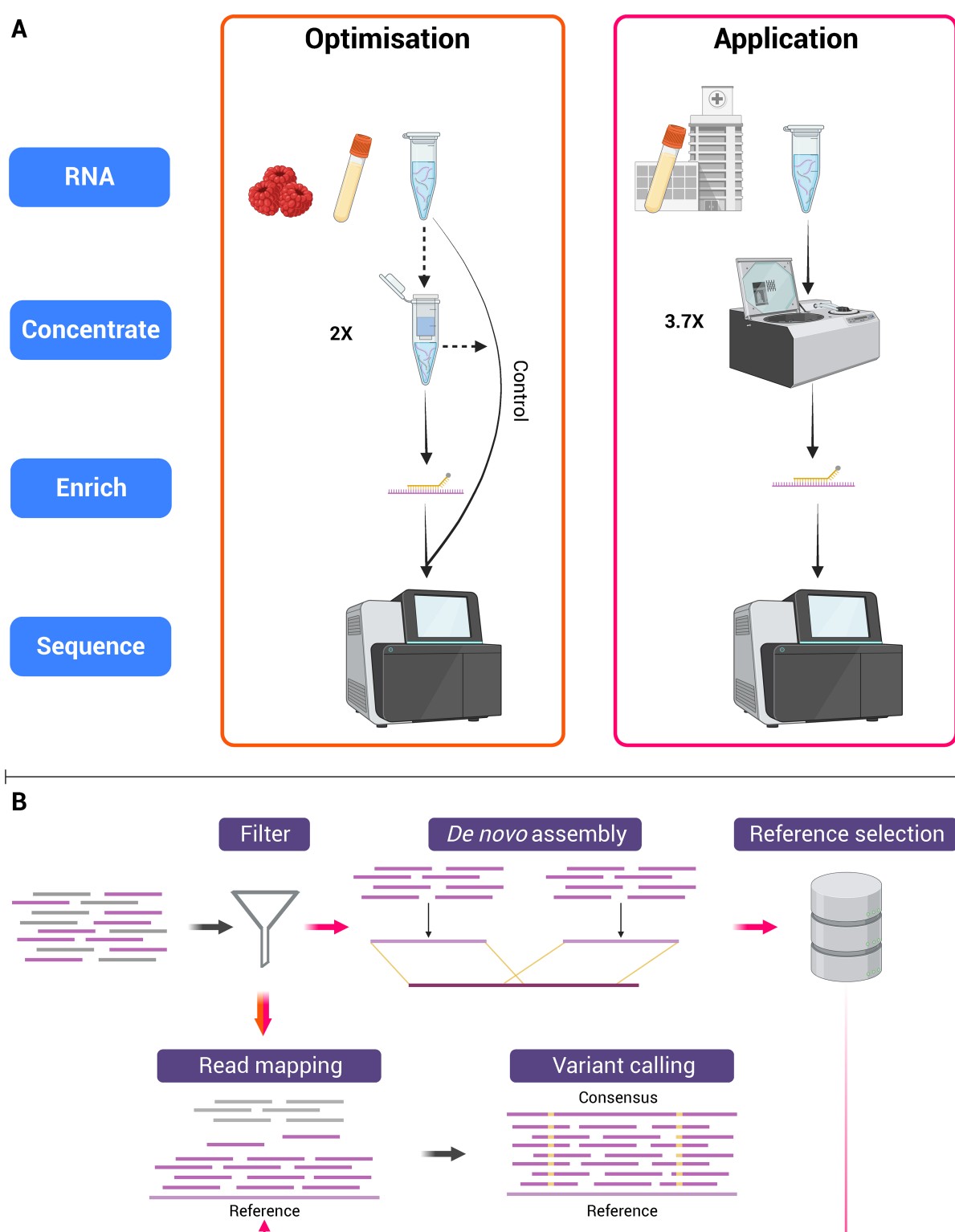

**FIG 1** Diagram illustrating laboratory (A) and bioinformatic (B) workflows. (A) Two experiments, Optimization and Application, were conducted involving steps such as RNA extraction from berry and/or serum samples, concentration (either column-based or vacuum), enrichment, and sequencing. Dashed arrows indicate optional steps where replicates were performed with or without the variable. (B) In the Optimization phase, reads (shown in orange) underwent PCR duplicate filtering, mapping to a control reference, and variant calling. For the Application phase, reads (shown in pink) were filtered for PCR duplicates and target homology, followed by *de novo* assembly for reference selection. These filtered reads were then mapped to the chosen reference for variant calling. Diagram created using BioRender.com.

normalized to 200 ng and pooled for three-plex hybridization overnight (16 hours) using the previously described pan-HAV probes (100 uM). Final sequencing libraries were normalized to 20 nM prior to pooling for sequencing on an Illumina NextSeq2000 using a P1 2 × 150 bp reagent kit.

### Analysis of pre- and post-enrichment libraries

Reads were adapter trimmed using cutadapt (v.2.6) with parameters -a CTGTCTCTTATA CACATCT -A CTGTCTCTTATACACATCT for paired-end trimming of universal dual indices (36). Taxonomic classification of trimmed reads was performed with Kraken 2 (v.2.1.2) using the viral RefSeq database indexed on 5 May 2021 (available from https://ben-langmead.github.io/aws-indexes/k2) (37). Trimmed reads were mapped to the spike-in material reference sequence (GenBank accession M59809.1) using minimap2 (v.2.18-r1015) with parameters -ax sr (38, 39). PCR duplicates were removed from the mapped reads using samtools *markdup* (v.1.17) (40). Mapping and quality metrics were obtained from samtools *samstats* and qualimap *bamqc* (v.2.2.2a) and collated with multiQC (v.1.14) (41, 42). An overview of the bioinformatic workflows is illustrated in Fig. 1B.

## Application to clinical specimen

### Patient samples

Archived serum samples, collected in Victoria, AUS between February and May 2023, from two infected patients, were referred to the Microbiological Diagnostic Unit Public Health Laboratory for HAV WGS. A single patient had two samples taken 1 month apart (AUSMDU00085964 and AUSMDU00085966). AUSMDU00085964 was referred with subgenotype documented as IIIA, but no subgenotype data were provided for the second referral (AUSMDU00085965). Ethical approval was granted from the University of Melbourne Human Research Ethics Committee (HREC no. 2021-22158-20072-2).

### Sample preparation and sequencing

RNA extraction and quantification were performed as described earlier for serum samples. Prior to library preparation, RNA extracts were vacuum concentrated 3.7-fold for 26 minutes at ambient temperature. Library preparation was performed as described earlier in duplicate with ~7.5 µL input RNA. Total RNA was below the limit of quantification by Qubit HS RNA assay. Libraries were normalized to 4 nM prior to pooling for sequencing on an Illumina iSeq 100 using an i1 2 × 150 bp PE reagent v2 kit.

### Analysis of clinical sequences

Reads were quality controlled as described earlier. To determine the subgenotype of AUSMDU00085965, deduplicated reads were filtered to discard those that were taxonomically unclassified. Filtered reads were *de novo* assembled using SPAdes (v.3.15.5) with the rnaviral parameter enabled (43). The longest contig was queried against a local nt database of all HAV sequences available as of May 2023 from GenBank (taxon:12092) and HAVNet using BLAST (v.2.14.0) with an *e*-value threshold of $10^{-4}$ (44, 45). The highest-scoring matches belonged to subgenotype IIIA. Deduplicated reads for all samples were then mapped to the most recently available complete IIIA genome (MN062167.1) collected in 2019 from California, USA (17). Variants were called using samtools *mpileup* and iVar *consensus* (v.1.0) with parameters -m 10 -q 30 -t 0.9 (46). Mapping and quality statistics were obtained as before. Duplicate sample reads were concatenated to create high-quality consensus genomes, annotated using VADR (v.1.3), and deposited to GenBank (47).

Two sets of context sequences were selected for phylogenetic analysis, which were obtained by querying the individual whole genomes (*WG-query*) and the extracted VP1/2A region (*VP1/2A-query*) sequences against the local nt HAV database to retrieve the 50 highest-scoring matches for each patient. These highest matching

sequences to each patient were combined into a single WG-query or VP1/2A-query multifasta file, and duplicate sequences were removed. Three alignments were created using MAFFT (v.7.520): WGS consensus sequences with WG-query sequences, VP1/2A consensus sequences with VP1/2A-query sequences, and WGS consensus sequences with VP1/2A-query sequences (48). Maximum-likelihood phylogenetic tree construction was performed with IQtree (v.2.2.2.3) using the best-fit model and 1,000 bootstrap replicates (49).

### *In silico* probe efficiency assessment

Complete and near-complete (>95%; >7,051 nts) genomes of HAV from human hosts were downloaded from GenBank on 24 April 2023 ($N$ = 239). Sequences used to create the probe panel were removed, with the exception of the single available IIB genome (AY644670.1), and the remaining sequences were randomly subsampled to account for oversampling from outbreak studies by selecting no more than three sequences per country and year for each subgenotype using augur *filter grep* (v.21.1.0; $N$ = 86) plus the spike-in control sequence (50). ProbeTools (v.0.1.9) modules *capture* and *stats* were used to align the pan-HAV probe sequences against the selected target sequences to assess probe coverage and depth (51). ProbeTools *getlowcov* was used to find low-coverage regions, defined as 120-kmers with zero probe coverage at more than 40 consecutive sites. A Mann-Whitney $U$ test was performed to evaluate whether the low probe coverage regions identified in the control sequence had significantly lower read depth compared to the rest of the genome in enriched samples using scipy (v.1.18.1) (52).

## RESULTS

### Pan-HAV probe design

A total of 425, 120-mer biotinylated oligonucleotide probes were selected to achieve a minimum 1× tiling across the representative target genomes (Tables S1-2).

### Optimization experiment

Frozen berries and serum spiked with "high," "mid," and "low" viral genome titers of HAV RNA corresponded to Cτs 35.1, 40.1, and Undetermined (<LOD) for berries, and 33.6, 36.6, and 41.0 for serum, respectively (Table 1, Table S4). In berries, between 0 and 5 unique on-target reads were detected in the unenriched samples, yielding no genome coverage with a minimum read depth of 10×. In serum, between 69 and 346 unique on-target reads were detected in unenriched samples, yielding genome coverage ranging from 0% to 1.9% with a minimum read depth of 10.

An increase in read depth and genome coverage was observed in all enriched samples. After the enrichment of berry libraries, mean-fold enrichment increased the read depth by 315.2× in high-titer samples with concentration treatment and 21.4× without; by 3.5× in mid-titer samples with concentration and 7.3× without; and by 3.6× in low-titer samples with concentration and 1.3× without (Fig. 2A). Up to 74%, 2.3%, and <0.1% of the genome was recovered from high, mid, and low titers at a minimum read depth of 10, respectively. Target reads were present at ≥1 depth up to 99.5%, 81.8%, and 11.7% of the genome positions in high-, mid-, and low-titer berry samples, respectively (Fig. 2B). After the enrichment of serum libraries, mean-fold enrichment increased the read depth by 368.9× in high-titer samples with concentration and 662.3× without; by 109.5× in mid-titer samples with concentration and 23.0× without; and by 3.1× in low-titer-samples with concentration and 3.0× without (Fig. 2A). Up to 100%, 80.8%, and 10.3% of the genome was recovered from high, mid, and low titers using a minimum read depth of 10, respectively. Target reads were present at ≥1 depth up to 100%, 100%, and 91.4% of positions in high-, mid-, and low-titer serum samples, respectively (Fig. 2B).

**TABLE 1** Summary of sample variables and genome coverage results[a].

| Sample | Matrix | Cτ | Copy no. (copies/µL) | Concentrated | Unenriched | | Enriched | |
|---|---|---|---|---|---|---|---|---|
| | | | | | Mean RD | Coverage | Mean RD | Coverage |
| BerryLow | Berry | Undet. | Undet. | N | 0.0 | 0.0 | 0.2 | 0.0 |
| | | | | Y[†] | 0.0 | 0.0 | 0.2 | 0.0 |
| BerryMid | | 40.2 | 0.45 | N | 0.0 | 0.0 | 2.6 | 1.1 |
| | | | | Y[†] | 0.0 | 0.0 | 2.7 | 2.3 |
| BerryHigh | | 35.1 | 11.2 | N | 0.0 | 0.0 | 12.0 | 60.5 |
| | | | | Y[†] | 0.1 | 0.0 | 14.8 | 74.1 |
| SerumLow | Serum | 41.0 | 0.28 | N | 3.1 | 1.9 | 1.2 | 1.0 |
| | | | | Y[†] | 2.5 | 0.9 | 12.5 | 10.3 |
| SerumMid | | 36.6 | 4.4 | N | 2.1 | 0.0 | 19.2 | 77.7 |
| | | | | Y[†] | 2.0 | 0.0 | 28.0 | 80.8 |
| SerumHigh | | 33.6 | 27.9 | N | 1.2 | 0.0 | 324.7 | 100.0 |
| | | | | Y[†] | 2.3 | 0.6 | 160.7 | 99.6 |
| AUSMDU00085964-1 | Serum | 35.8 | 7.1 | Y[‡] | – | – | 2455.3 | 100.0 |
| AUSMDU00085964-2 | | | | Y[‡] | * | * | * | * |
| AUSMDU00085966-1 | | 42.7 | 0.09 | Y[‡] | – | – | 24.3 | 93.3 |
| AUSMDU00085966-2 | | | | Y[‡] | – | – | 23.0 | 89.5 |
| AUSMDU00085965-1 | | 34.7 | 14.07 | Y[‡] | – | – | 3446.4 | 100.0 |
| AUSMDU00085965-2 | | | | Y[‡] | – | – | 2387.9 | 100.0 |

[a]Undetermined, Undet. Cτ values occur when the target is not present or is present below the LOD of the assay used. Copy number is the genome copies/µL quantified from the RNA extract as determined by the ISO 15216–1 HAV quantification real-time RT-PCR protocol. Mean read depth (RD) is calculated as the average number of deduplicated reads covering one position in the genome. Coverage reported here is the percentage of genome positions with a depth of 10 or more reads. A summary of positive and negative controls can be found in Table S4.

[b] †, Column-based concentration with DNase treatment.

[c] ‡ , Vacuum concentration.

## Patient samples

One patient sampled twice over a 1-month period had a Cτ value of 35.8 for the first (AUSMDU00085964) and 42.7 for the second (AUSMDU00085966). AUSMDU00085965 had a Cτ value of 34.7 (Table 1, Table S4). Genotyping was available for AUSMDU00085964, thus presumed for AUSMDU00085966, reported as IIIA. The genotype for AUSMDU00085965 was determined to be IIIA using the longest *de novo* assembled contig (3,482 nt) queried against the local HAV database. Enriched libraries of AUSMDU00085964, AUSMDU00085966, and AUSMDU00085965 were composed of up to 12.9%, 17.2%, and 0.9% HAV classified reads (Fig. 3A), yielding a mean depth up to 2455.3×, 24.3×, and 3446.4× unique target reads (Table 1), respectively. The complete genome (100%) was recovered for AUSMDU00085964 and both AUSMDU00085965 replicates, while 89.5% and 93.3% were recovered for AUSMDU00085966 replicates using a minimum read depth of 10. Target reads were present at ≥99.9% of genome positions in all samples (Fig. 3B). Besides ambiguous bases, replicate consensus genomes were highly concordant with one discordant SNV in AUSMDU00085966.

Comparing the nucleotide differences in the WG and VP1/2A region (503 nt) between the complete sequences of both patients, there are 80 (0.011 substitutions/site) and 6 SNVs (0.012 substitutions/site), respectively. All 51 of the context sequences obtained by querying the WG sequences were >7 kb, while 71.2% of the 52 context sequences obtained by querying the VP1/2A sequences were <1 kb. Phylogenetic analysis of the WG sequences with WG-query context samples places the patient samples in the most recent clade descending from a clade of strains collected from India in the early 2000s (Fig. 4A). Patient samples cluster together in the tree of VP1/2A sequences with VP1/2A-query context samples amongst contemporary sequences, primarily collected in European countries (Fig. 4B). In the tree comparing WG sequences with VP1/2A-query context samples, patient samples are similarly found in a contemporary European clade but are distributed further apart (Fig. 4C).

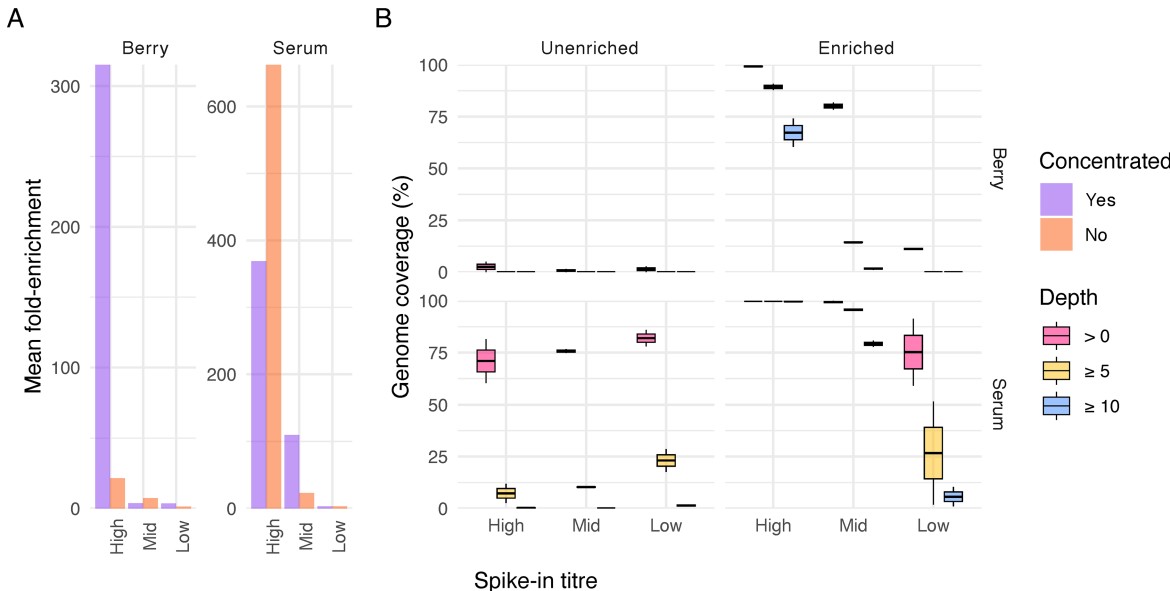

**FIG 2** (A) Mean-fold enrichment between paired pre- and post-enrichment samples. To avoid infinite values or overestimation using a small pseudocount, a depth of 1 was assigned to positions in pre-enriched samples where the corresponding position in the post-enriched sample was greater than 0 under the biological assumption that one or more reads must have been present for enrichment to have occurred. (B) Genome coverage by spike-in titer levels at read depth thresholds of >0, ≥5, and ≥10.

### *In silico* prediction

A table of accession numbers and metadata for sequence data used for *in silico* prediction are available in Table S3. On average, 98.6% of the queried target positions were covered by one or more probes (median 99.2, IQR 98.8–100). Subgenotypes I—IIIA and IIB are covered by one or more probes at >99.1% positions, while IB and IIIB have 97.7% and 92.8% coverage, respectively (Fig. 5). Amongst strains with more than one target sequence, an average of 1.1 low probe coverage regions were identified. The single IIB and IIIB strains had zero and five low-probe coverage regions, respectively. Three low-probe coverage sites were predicted in the control sequence between positions 1,690–1,809, 2,633–2,752, and 6,730–6,849, none of which had significantly lower read depth compared to the rest of the genome.

### DISCUSSION

We present a hybrid capture oligo panel for WGS of all six human pathogenic HAV subgenotypes from serum and food samples, demonstrating a highly sensitive and generalized HAV WGS workflow for outbreak surveillance. This approach is a more cost-effective alternative to HAV WGS by metagenomic sequencing, can be used with degraded or challenging sample matrices, and can be employed without *a priori* genotyping.

### Optimization experiment

The enrichment of the complete HAV genome from berry specimen was observed as low as 83 input copies and near complete (78.3%) from four input copies, providing the most cost-effective approach of its kind to date. Previously, deep metagenomic sequencing had recovered a near-complete HAV from frozen raspberries, generating 5M reads and yielding a mean read depth of 60 (54). Lower HAV read depth observed in berry specimen compared to serum may be attributed to lower genome input copies and an over-fragmented library. To address the former, future optimization should increase the input volume into the concentration workflow to both remove residual inhibitors and increase input viral RNA. Regarding the latter, the vigorous purification process

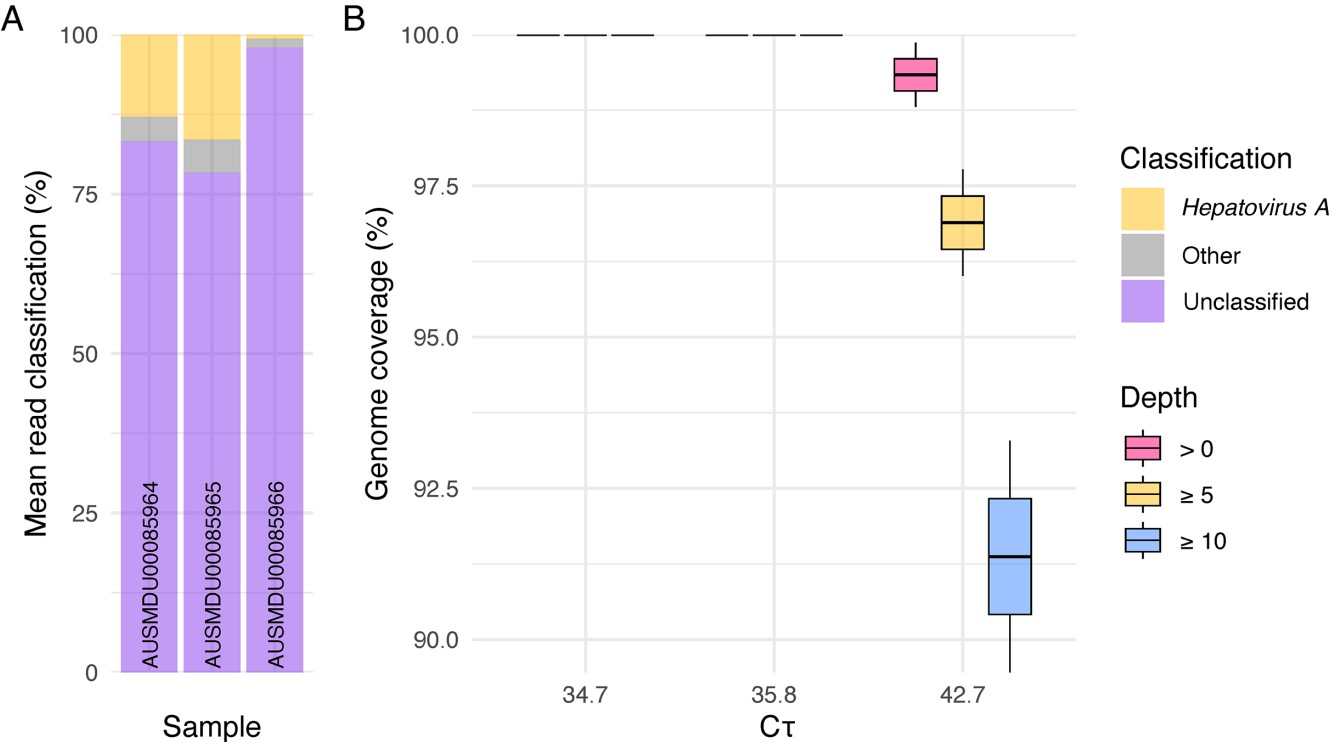

**FIG 3** HAV read statistics of enriched clinical specimen. (A) Taxonomic classification of sample reads averaged over technical duplicates belonging to *Hepatovirus A*, non-HAV viral taxonomies, and unclassified. (B) Box and whisker plot of genome coverage by sample Cτ at read depth thresholds of >0, ≥5, and ≥10.

necessary for removing inhibitors from food samples resulted in nearly all sequencing libraries having average read lengths <120 (Fig. S1). Previous work has reported reduced hybridization efficiency when the mean insert size of the library was shorter than the probe length (55). Another possibility is residual inhibitors present in the berry RNA extracts could negatively affect cDNA synthesis and tagmentation prior to hybridization, which could explain why mean-fold enrichment of cleaned and concentrated berry samples was significantly greater than the untreated berry samples at high titer. Whereas the inverse was observed in lower-titers, and serum, suggesting sample loss in the spin-column outweighs the benefits of purification (Fig. 2A). It is recommended that future optimization includes concentration from higher input volumes to increase viral RNA input and further determination of titers suitable for column-based purification for food or other complex matrices. Additional troubleshooting is needed to minimize over fragmentation during library tagmentation.

In serum samples, the complete genome was recovered from as few as 65 input copies and near-complete (91.4%) from 4 input copies. These results are similarly sensitive compared to amplicon-based sequencing for HAV WGS from serum but offer a simplified solution for routine sequencing in public health laboratories. For instance, the IA scheme presented by Lee et al. yielded a maximum of 94.7% genome coverage in specimen with Cτs ranging from 27.6 to 33.3. Cleary et al. designed a scheme for IA-B where nearly half of the samples with Cτ values between 16 and 40 yielded ≥95.0% genome coverage using a combination of multiplex amplicon and Sanger sequencing (55). Previous iterations of this scheme included IIA resulting in lower amplification efficiency due to the degree of genetic variation between the genotypes. Unlike primer schemes for amplicon-based sequencing, hybrid capture allows for the enrichment of all six subgenotypes using a pan-HAV probe panel. Moreover, hybrid capture can tolerate thousands of probes, meaning additional targets can be added to the current panel to facilitate robust pathogen surveillance.

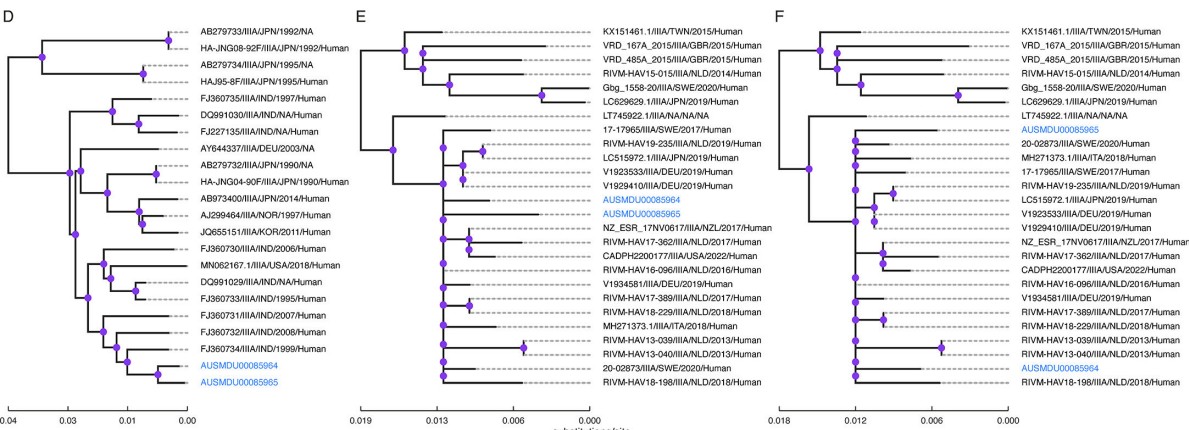

**FIG 4** Maximum likelihood trees comparing WGS- and VP1/2A-query results. Ancestral nodes were traversed to clearly visualize patient samples (blue). Nodes with high bootstrap support (≥80%) are colored in purple. Trees were visualized using toytree [v.2.0.5; (53)]. (A) WGS consensus sequences in the context of WG-query sequences (best-fit model = TIM2 + F + I + R3). (B) VP1/2A consensus sequences in the context of VP1/2A-query sequences (best-fit model = TIM2 + F + I + G4). (C) WGS consensus sequences in the context of VP1/2A-query sequences (best-fit model = TIM2 + F + I + G4).

## Patient samples

After observing potential target loss from column-based concentration in the optimization experiment, vacuum concentration was implemented with the clinical specimen as well as using a greater RNA extract input volume (30 µL) prior to enrichment. As a result, mean read depth and genome coverage from patient samples increased compared to contrived samples with similar Cτ values. At a minimum read depth of 10, complete genomes were recovered from AUSMDU00085964 and AUSMDU00085965 and near complete from AUSMDU00085966 (93.3%). However, 99.9% of the positions in AUSMDU00085966 were covered by one or more reads, suggesting the possibility of recovering a complete genome with greater input volume or sequencing depth. Indeed, combining reads from technical duplicates yields 97.7% coverage and 46.7 mean read depth. Regardless, such recovery at this Cτ is remarkable given the difficulty of detecting subgenotype IIIA. Recently, a reverse transcription-PCR (rRT-PCR) HAV subgenotyping assay found the LOD for IIIA to be 10 RNA copies per reaction, equivalent to 34.2 and 33.9 Cτ for singleplex and triplex reactions, respectively (56). Whereas a subgenotype-specific optimized reverse transcription quantitative real-time PCR (RT-qPCR) assay found the LOD for IIIA to be 5,000 copies per reaction, equivalent to Cτ near 41.3 (57). This may explain why subgenotyping data were not available for AUSMDU00085965, or typing had not been requested. Given the sensitivity observed in this study, this assay could be useful for investigating false negative cases with corroborating clinical or epidemiological evidence.

It has been demonstrated that the whole genome can provide greater geographic and temporal resolution for virus phylogenetics, generally (28). For HAV, specifically, WGS has been shown to increase phylogenetic resolution and provide additional context in the absence of epidemiological data (27, 29). Even though substitutions per site were slightly higher in the VP1/2A region compared to the whole genome sequence (0.012 vs 0.011), phylogenetic analysis comparing whole genomes to the top VP1/2A hits provided additional resolution due to the sequences in the alignment including additional regions around the VP1/2A junction (Fig. 4B and C). The previous difficulty of WGS HAV is reflected in the sequence database where few complete genomes are available, thus resulting in high similarity to sequences more than 15 years old (Fig. 4A). Until HAV genomes are better represented in these databases, it is recommended that subgenomic regions are used to select closely related sequences for phylogenetic analysis of the whole genome.

## *In silico* prediction

We observed high-capture efficiency in two of the six subgenotypes *in vitro*. Similarly, *in silico* prediction shows broad specificity across all human subgenotypes with one or more probes covered at >92% genome positions (Fig. 5). Analysis of low-probe coverage found an average of 1.1 120-kmer regions, representing less than 2% of the HAV genome. While low-probe coverage regions were identified *in silico*, variables such as random fragmentation and hybridization temperature can improve efficiency *in vitro*. In fact, we did not observe significantly lower read depth in regions of the enriched contrived samples where low-probe coverage was predicted in the control sequence.

## Limitations

This study serves as a proof-of-concept for the feasibility of a highly sensitive culture-agnostic method for HAV WGS; however, additional work is needed to further optimize and validate the approach for implementation in public health. For example, more complex sample types of public health relevance remain untested, such as stool, wastewater, or shellfish. Promising results have been found from ultra-deep sequencing of wastewater enriched with a broad viral discovery probe panel, yielding three partial HAV genomes (1,482–6,729 nts) (58, 59). Cost may also be a limiting factor for wider implementation. In the present study, the price per sample was $157AUD, approximately three-times the cost of highly multiplexed amplicon sequencing (60). However, increased multiplexing as well as additional optimization to increase target read proportions could lower per-sample costs. Furthermore, we anticipate the value of this approach to be

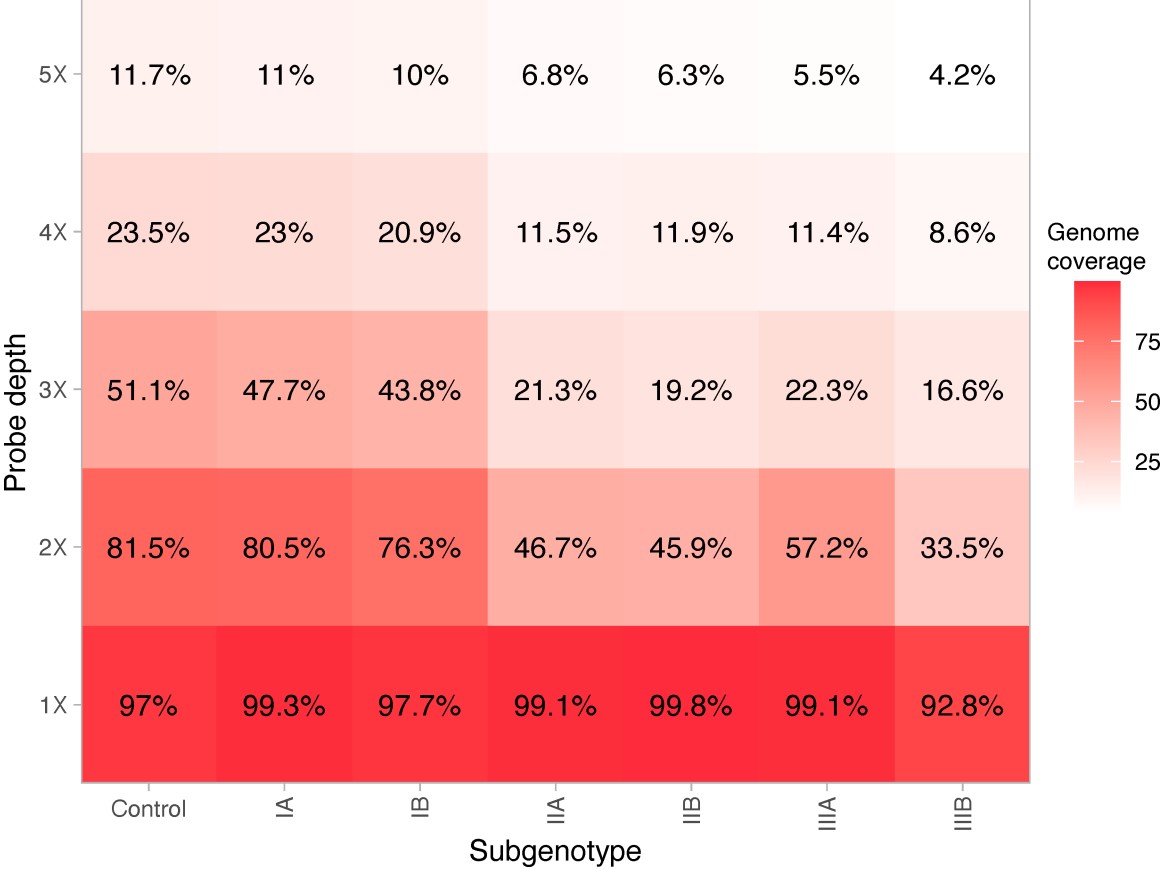

**FIG 5** *In silico* prediction of probe efficiency, averaged by subgenotype and stratified by probe coverage depth. Percentage annotation indicates the mean predicted percent genome coverage at respective probe depths. Subgenotypes IIB and IIIB each have a single genome available for comparison (AB279735.1 and AY644670.1, respectively).

the greatest for supporting food-borne surveillance, where PCR-based amplification is limited by low, degraded viral input and the presence of inhibitors (61, 62). For food specimen in particular, optimization should evaluate the effect of pH and organic load, as well as nucleic acid shearing during virus extraction, on enrichment. Finally, overnight hybridization was used to accommodate hands-on staffing schedules. Implementation of liquid handing automation could bring the turnaround time from extraction to sequencer to under 24 hours.

## Conclusion

This study presents, for the first time, a solitary approach for achieving highly sensitive WGS of HAV from clinical and food samples, while proving relatively cost-effective. WGS of HAV can support outbreak investigations by offering enhanced phylogenetic resolution and simultaneously facilitate the refinement and optimization of molecular diagnostic assays through the deposition of genome data in public databases. In conclusion, flexible oligo design and hybridization-based enrichment make this a promising workflow for streamlining the surveillance of enteric viruses, thereby furthering the genomic capacities of public health laboratories.

## ACKNOWLEDGMENTS

This work was supported by the National Health and Medical Research Council, Australia Partnership Grant (GNT1149991), and the Australian Government Medical Research Future Fund, Genomics Health Futures Mission Flagships—Pathogen Genomics Grant (FSPGN000045). S.E.Z. is supported by a Melbourne Research Scholarship from the University of Melbourne.

We are grateful to Dr. Andrew Gador-Whyte and the Department of Microbiology at Austin Health for their assistance. We thank the Victorian Infectious Disease Reference Laboratory for their valuable guidance in the conceptualization of this project. We would also like to show our gratitude to Dr. Ketan Patel for insightful comments on the manuscript.

## AUTHOR AFFILIATIONS

[1]The Center for Pathogen Genomics, The University of Melbourne, Melbourne, Victoria, Australia
[2]Department of Microbiology and Immunology, The University of Melbourne at the Peter Doherty Institute for Infection and Immunity, Melbourne, Victoria, Australia
[3]Microbiological Diagnostic Unit Public Health Laboratory, The University of Melbourne at the Peter Doherty Institute for Infection and Immunity, Melbourne, Victoria, Australia
[4]Department of Infectious Diseases, Austin Health, Heidelberg, Victoria, Australia
[5]Department of Infectious Diseases, The University of Melbourne at the Peter Doherty Institute for Infection and Immunity, Melbourne, Victoria, Australia
[6]Doherty Applied Microbial Genomics, The University of Melbourne at the Peter Doherty Institute for Infection and Immunity, Melbourne, Victoria, Australia

## AUTHOR ORCIDs

Sara E. Zufan  http://orcid.org/0000-0002-3606-0297
Timothy P. Stinear  http://orcid.org/0000-0003-0150-123X

## FUNDING

| Funder | Grant(s) | Author(s) |
| --- | --- | --- |
| DHAC | National Health and Medical Research Council (NHMRC) | GNT1149991 | Benjamin P. Howden |
| | | Torsten Seemann |

| Funder | Grant(s) | Author(s) |
| --- | --- | --- |
| | | Timothy P. Stinear |
| Australian Government Medical Research Future Fund | FSPGN000045 | Jason C. Kwong |
| | | Benjamin P. Howden |
| | | Torsten Seemann |
| | | Timothy P. Stinear |

## AUTHOR CONTRIBUTIONS

Sara E. Zufan, Conceptualization, Data curation, Formal analysis, Investigation, Methodology, Visualization, Writing – original draft, Writing – review and editing | Karolina Mercoulia, Conceptualization, Data curation, Resources, Writing – review and editing | Jason C. Kwong, Resources, Supervision, Writing – review and editing | Louise M. Judd, Data curation, Methodology, Project administration, Writing – review and editing | Benjamin P. Howden, Funding acquisition, Supervision, Writing – review and editing | Torsten Seemann, Supervision, Writing – review and editing | Timothy P. Stinear, Conceptualization, Funding acquisition, Supervision, Writing – review and editing

## DATA AVAILABILITY

Complete genome sequences from AUSMDU00085964 and AUSMDU00085965 were deposited in GenBank (accession numbers OR261026.1 and OR261027.1). Supplementary materials, including sequencing reads from the optimization experiment, are available from https://doi.org/10.26188/23648025.

## ADDITIONAL FILES

The following material is available online.

### Supplemental Material

**Fig. S1 (Spectrum02834-23-S0001.tiff).** Read length distributions.
**Supplemental legends (Spectrum02834-23-S0002.docx).** Legends for supplemental tables and figure.
**Supplemental tables (Spectrum02834-23-S0003.xlsx).** Tables S1 to S4.

### Open Peer Review

**PEER REVIEW HISTORY (review-history.pdf).** An accounting of the reviewer comments and feedback.

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
