## [Reviewer comments · Microbiology Spectrum]

Microbiology Spectrum

High performance enrichment-based genome sequencing to support the investigation of hepatitis A virus outbreaks

Sara Zufan, Karolina Mercoulia, Jason Kwong, Louise Judd, Benjamin Howden, Torsten Seemann, and Timothy Stinear

Corresponding Author(s): Sara Zufan, The University of Melbourne

Review Timeline:

Submission Date:	July 24, 2023
Editorial Decision:	September 15, 2023
Revision Received:	September 27, 2023
Accepted:	October 14, 2023

Editor: Kalliopi Rantsiou

Reviewer(s): The reviewers have opted to remain anonymous.

Transaction Report:

DOI: <https://doi.org/10.1128/spectrum.02834-23>

September 15, 2023

Ms. Sara E Zufan
The University of Melbourne
Department of Microbiology and Immunology
Melbourne
Australia

Re: Spectrum02834-23 (High performance enrichment-based genome sequencing to support the investigation of hepatitis A virus outbreaks)

Dear Ms. Sara E Zufan:

Link Not Available

Sincerely,

Kalliopi Rantsiou

Journals Department
Reviewer comments:

Reviewer #1 (Comments for the Author):

The authors do a nice job in describing the need for and justification of the proof-of-concept system. It is great that the authors chose frozen berries as their matrix. Identification of clusters and their origin-based o sequence is important.

The detection system is described in the abstract as being used with human serum samples of high CT values. The authors should clarify why CT values of this high are being seen as positive and issues with this. Would this also translate for use then with environmental samples of greater complexity of inhibitors but with low levels of virus and high CT values? The Importance statement includes a sentence about challenging samples but does not mention environmental samples. Can the authors

speculate about the application to environmental samples?

What normally happens with a sample of a CT value of 38 or >40? Is this normally seen as positive?

Is there a reference for use of the LENTICULE discs?

Since GenBank contains many partial sequences of HAV, is it of use to note the parameters used to download the chosen sequences from GenBank?

Line 289 should be cost- effective approach.

In lines 291-292 the differences in berry and serum samples are discussed. What is the impact of pH and organic load? Was the pH and other characteristic measured as they would be important in further enhancing the use of this tool.

The raw data in the supplemental files is likely to be useful to some researchers.

Reviewer #2 (Comments for the Author):

The authors developed an oligonucleotide hybrid capture enrichment-based sequencing method to detect the hepatitis A virus. They demonstrated this method is highly sensitive and could be promising for future hepatitis A virus outbreak surveillance. There are some problems that should be addressed.

Only phylogeny was carried out and genomic analysis was not sufficient for the comparison between the near-complete genomes and their reference genomes to see how different or similar they were. The authors should perform more genomic analysis for viruses (e.g. genome annotation) to look into the data at a deeper level. It is more important to see how the near-complete genomes performed in downstream genomic analysis commonly performed for viral genomes.

Any reason why different sequencing platforms (iSeq and NextSeq) were used for the optimization experiment and clinical samples?

It would be helpful if you could provide a flow chart to display your workflow, including experiments and subsequent bioinformatic analysis.

Did you test the difference between using SPAdes and metaSPAdes for the filtered reads?

Lines 117 and 119: used

Lines 137 and 139: 222 µL...200 µL. A space should be added before µL throughout the manuscript.

Staff Comments:

Preparing Revision Guidelines

Please return the manuscript within 60 days; if you cannot complete the modification within this time period, please contact me. If you do not wish to modify the manuscript and prefer to submit it to another journal, please notify me of your decision immediately so that the manuscript may be formally withdrawn from consideration by Microbiology Spectrum.

Reviewer #1

1. The authors should clarify why CT values of this high are being seen as positive and issues with this. Would this also translate for use then with environmental samples of greater complexity of inhibitors but with low levels of virus and high CT values?

A. High Ct values (including values >40) as they relate to different HAV genotypes LOD is alluded to in L48-50 and is based on previous research we cite in the manuscript (reference 56). We speculate on the performance of hybrid capture in wastewater samples in the discussion (L361-364).

2. The Importance statement includes a sentence about challenging samples but does not mention environmental samples. Can the authors speculate about the application to environmental samples?

A. Added to L54-55; discussed in L360-362.

3. What normally happens with a sample of a CT value of 38 or >40? Is this normally seen as positive?

A. At Ct of 38 is approximately equivalent to one genome copy per uL. According to the ISO 15216 validation documentation, the LOD₅₀ for HAV in raspberries is 0.92 (0.41 to 2.04). In the applied sense, positive detection >38 Ct would be considered a spurious result. The implications of high Cts are discussed in L344-351. Additional context of the significance of this was added to L351-353.

4. Is there a reference for use of the LENTICULE discs?

A. Yes, it is provided in L165.

5. Since GenBank contains many partial sequences of HAV, is it of use to note the parameters used to download the chosen sequences from GenBank?

A. The database contains all sequences, regardless of length. This has been clarified in L191-192.

6. Line 289 should be cost- effective approach.

A. Corrected

7. In lines 291-292 the differences in berry and serum samples are discussed. What is the impact of pH and organic load? Was the pH and other characteristic measured as they would be important in further enhancing the use of this tool.

A. Per ISO 15216-1:2017 protocol for soft fruit, pH is adjusted to 7.0 +/- 0.5 during the virus extraction step. It is possible some remaining inhibitors are present that could

inhibit cDNA synthesis, supported by significantly higher enrichment in high-HAV-titre berry samples that underwent the RNA clean and concentrate step compared to those samples that did not (Figure 1A). However, the neutralisation process is possibly the cause of sheared nucleic acids observed in berry samples (Figure S1), which would reduce enrichment efficiency. We are planning to seek national laboratory testing accreditation for this method which will consider both inhibitors and NA shearing in various food samples. This context has been added to L368-369.

8. The raw data in the supplemental files is likely to be useful to some researchers.

A. The DOI has been provided L224.

Reviewer #2

1. Only phylogeny was carried out and genomic analysis was not sufficient for the comparison between the near-complete genomes and their reference genomes to see how different or similar they were. The authors should perform more genomic analysis for viruses (e.g. genome annotation) to look into the data at a deeper level.
 - A. The genomes were annotated prior to submission to Genbank, this has been added at L200. Nucleotide comparisons between replicates found near 100% concordance, where most discrepancies were the result of low depth and stringent variant calling threshold. This is now described in L270-271. We agree that more can be learned from deeper genomic analyses. However, they would be beyond the scope of application in a public health laboratory at this time, especially for HAV where outbreak management has been limited to genotypic characterization. We hope this assay will allow others to conduct genomic studies to improve our knowledge of HAV pan genome and structural differences.

2. It is more important to see how the near-complete genomes performed in downstream genomic analysis commonly performed for viral genomes.
 - A. We agree. A number of previous HAV studies have shown that the whole genome provides additional resolution compared to some genes commonly used for phylogenetic analysis. Similarly, it has been observed with SARS-CoV-2 that genome coverage <90% results in altered branch length and lineage assignment compared to its complete replicate. We hope to use this manuscript to solicit collaborations where we can obtain more samples with epidemiological data to complete more robust analyses to better characterise the effect of missing nucleotide data on the genomic epidemiology of HAV.

3. Any reason why different sequencing platforms (iSeq and NextSeq) were used for the optimization experiment and clinical samples?
 - A. No, we used what was available at the time. However, reads were normalised to 1M per index to account for differences in sequence abundance.

4. It would be helpful if you could provide a flow chart to display your workflow, including experiments and subsequent bioinformatic analysis.

A. Now included as Figure 1.

5. Did you test the difference between using SPAdes and metaSPAdes for the filtered reads?

A. No. We did, however, implement metaSPAdes on the full read set. Given the high Ct-value of the samples, most reads (Figure 3A) were off-target, it was computationally faster to filter reads using Kraken than running metaSPAdes. We will make this comparison when validating the bioinformatic workflow in the next phase of the project.

6. Lines 117 and 119: used

A. Corrected.

7. Lines 137 and 139: 222 μ L...200 μ L. A space should be added before μ L throughout the manuscript.

A. Corrected throughout.

October 6, 2023

Ms. Sara E Zufan
The University of Melbourne
Department of Microbiology and Immunology
Melbourne
Australia

Re: Spectrum02834-23R1 (High performance enrichment-based genome sequencing to support the investigation of hepatitis A virus outbreaks)

Dear Ms. Sara E Zufan:

Your manuscript has been accepted, and I am forwarding it to the ASM Journals Department for publication. You will be notified when your proofs are ready to be viewed.

Sincerely,

Kalliopi Rantsiou
Editor, Microbiology Spectrum
